# Perspectives of informal caregivers who support people following hip fracture surgery: a qualitative study embedded within the HIP HELPER feasibility trial

Allie Welsh ,[1] Sarah Hanson ,[1] Klaus Pfeiffer,[2] Reema Khoury,[3] Allan Clark ,[4] Polly-Anna Ashford,[3] Sally Hopewell ,[5] Pip Logan ,[6] Maria Crotty ,[7] Matthew Costa ,[5] Sallie Lamb,[8] Toby Smith ,[1,9] Collaborators HIP HELPER Study[1]

For numbered affiliations see end of article.

**Correspondence to**
Dr Allie Welsh;
a.welsh@uea.ac.uk

## ABSTRACT

**Objectives** This study aims to illuminate the perspectives of informal caregivers who support people following hip fracture surgery.

**Design** A qualitative study embedded within a now completed multicentre, feasibility randomised controlled trial (HIP HELPER).

**Setting** Five English National Health Service hospitals.

**Participants** We interviewed 20 participants (10 informal caregivers and 10 people with hip fracture), following hip fracture surgery. This included one male and nine females who experienced a hip fracture; and seven male and three female informal caregivers. The median age was 72.5 years (range: 65–96 years), 71.0 years (range: 43–81 years) for people with hip fracture and informal caregivers, respectively.

**Methods** Semistructured, virtual interviews were undertaken between November 2021 and March 2022, with caregiver dyads (person with hip fracture and their informal caregiver). Data were analysed thematically.

**Findings** We identified two main themes: expectations of the informal caregiver role and reality of being an informal caregiver; and subthemes: expectations of care and services; responsibility and advocacy; profile of people with hip fracture; decision to be a caregiver; transition from hospital to home.

**Conclusion** Findings suggest informal caregivers do not feel empowered to advocate for a person's recovery or navigate the care system, leading to increased and unnecessary stress, anxiety and frustration when supporting the person with hip fracture. We suggest that a tailored information giving on the recovery pathway, which is responsive to the caregiving population (ie, considering the needs of male, younger and more active informal caregivers and people with hip fracture) would smooth the transition from hospital to home.

**Trial registration number** ISRCTN13270387.Cite Now

## STRENGTHS AND LIMITATIONS OF THIS STUDY

⇒ We describe the contextual factors of participating in HIP HELPER and people's experiences of giving and receiving informal care.

⇒ This study may help health professionals support informal caregiver dyads in the transition from hospital to home.

⇒ Participants presented with limited socioeconomic, cultural or ethic perspectives, and there was a lack of perspectives of those caring for a person with cognitive impairment.

⇒ The COVID-19 pandemic affected National Health Service services, which may have impacted on the study delivery and participant's experiences.

80 000 people aged 60 years and over experience a hip fracture in the UK each year, resulting in a combined health and social cost of at least £2 billion.[2] This cost is expected to increase with an ageing population.[1] Outcomes following hip fracture are poor. The majority of individuals do not return to preinjury levels of function, and frequently lose independence and self-caring abilities.[3 4]

Given the sudden, traumatic nature of hip fracture, people often rely on family members or friends for informal care following hospital discharge. Additionally, approximately 40% of people who sustain a hip fracture have cognitive impairment, meaning many caregivers also have to cope with the challenges of functional dependency and neurocognitive symptoms.[5] These 'informal caregivers', therefore, play an integral role, in the transition from hospital to home, the initial discharge period and possibly beyond.[6]

Informal caregivers are a heterogeneous population. For some, this may be the first time with caregiving responsibilities; for others, it is perceived as an episodic event

## INTRODUCTION

Hip fracture is a devastating injury, predominantly seen in frail older people and disproportionally more women.[1] Approximately

with a temporary increase in care needs. They may be a retired spouse, friend, grown-up children in employment; living with, close by or far away; and of course, they are facing their own physical and mental health challenges. Caregiving may further be exacerbated by (but not limited to) the ageing population, smaller average family size, changing gender-roles due in part to more women working outside the home.[7–9]

The ability of a person to cope with the roles and responsibility of caregiving is a complex interaction of different factors. In addition to stressors related to the actual caregiving, there are other factors that may determine a person's ability to manage the caregiving role, such as caregiver resources, social environment (eg, informal and professional support), relationship quality with the care recipient and caregiver, positive aspects of caregiving (eg, finding meaning in the caregiving role).[10] While caregiving experiences have been reported in other populations such as stroke[6 11] and dementia,[12] there is still much to understand about acute episodes of informal care, such as in hip fracture. Previous literature suggests that the experiences of recovery after hip fracture for patients and informal caregivers may be different to these previous populations.[13] These may include differing recovery trajectories and impairment,[14] greater challenges with postoperative pain management, nausea and fear avoidance.[15]

Our qualitative study investigated the informal caregiver perspectives of caring for a person with hip fracture following hospital discharge, nested within a larger study aimed at assessing the feasibility of a pragmatic, multicentre randomised controlled trial (RCT) of an informal caregiver training programme (HIP HELPER) to support the recovery of people following hip fracture surgery.[16]

## METHODS
This was a qualitative study embedded within a multicentre, feasibility RCT (HIP HELPER). The HIP HELPER protocol is published elsewhere, and the study is now complete.[16] In brief, HIP HELPER aimed to assess the design of a pragmatic RCT to test the effectiveness of an informal caregiver intervention, compared with usual National Health Service care following hip fracture surgery. Caregiver dyads who were randomised to receive the HIP HELPER intervention were allocated to receive three, 1-hour, one-to-one training sessions. Training session had standardised content and, all of which were delivered by a nurse, a physiotherapist or an occupational therapist who had been trained to deliver the content. Training sessions included practical skills for rehabilitation such as transfers and walking, pacing, and stress management techniques, and the provision of and the HIP HELPER Caregiver Workbook, offered information on recovery, exercises, worksheets and goal-setting plans to facilitate a 'good' recovery.

The present study reports the experiences and contextual factors of undergoing hip fracture surgery and dyad's

experiences of giving and receiving informal care. Patient and caregiver perspective of the experimental caregiving training programme are to be reported with the feasibility study results. Group allocation (intervention or control) was, therefore, not a significant consideration for the present study. We followed the Consolidated Criteria for Reporting Qualitative Studies (COREQ) reporting guidelines.[17]

Fourteen caregiver dyads were purposively sampled by age, prefracture disability and hospital locations. Patients were aged 60 years and above, had undergone hip fracture surgery and had a nominated individual who acted as an informal caregiver. An informal caregiver was defined as someone who was expected to informally provide care, assistance, support or supervision in activities of daily living (ADLs) for at least 3 hours per week, but not on a paid basis. This may have included personal ADLs such as toileting, washing, dressing and eating; and/or more complex tasks such as managing money, shopping and household chores.[18 19] Other inclusion criteria for participants were community-dwelling prior to admission, able to attend face-to-face hospital appointments and/or access to a computer/table and internet services to receive a video consultation call.

## Data collection
In-depth, semistructured interviews were undertaken with caregiver dyads, within 6 weeks of hospital discharge, by AW, who is a white, female postdoctoral researcher. AW had no role in recruitment to the study nor intervention delivery, thus was not known to participants. Interviews were conducted virtually using Microsoft Teams or via telephone, between November 2021 and March 2022. They were audiorecorded, transcribed and all identifying information removed.

It was not necessary to operationalise the concept of saturation, firstly, given our sampling approach, and secondly, we recognise that meaning is generated through interpretation of data are inescapably situated and subjective.[20 21] As per Hanson *et al*'s[22] approach, we analysed the data to identify recurrent themes generated from the analysis. At 10 participant-dyads, we started to see recurring themes and ideas. Accordingly, we made a pragmatic decision that we had collected sufficient data to be representative of the hip fracture informal caregiver dyad experiences and ceased data collection.

## Data analysis
Using the principles of thematic analysis, data were initially analysed deductively against our questioning framework and then further explored inductively to explore contextual features and participants experiences.[23] AW generated initial codes and themes. AW, SHa, TS and KP then engaged in discussion to develop, review, refine and name themes.

Reflexivity was acknowledged in the design and analysis of our study. This included reflecting on the positions of researcher team (TS—physiotherapist, SHa—nurse,

AW—physiologist and KP—psychologist) to ensure we understood the context and significance were interpreted appropriately and our predetermined positions were appreciated.[24 25]

We did not return our transcripts to participants for comment or correction, and we did not seek ethical approval to do so. This is based on our experience as qualitative researchers and the little evidence available that suggests member checks improve research findings.[26 27] Participants were thus not involved in the data analysis process for these reasons and the lack of resources available for training individuals.

### Patient and public involvement
Patient involvement began during protocol development and continued throughout the trial. The patient member was a coinvestigator, who provided insights into the trial conduct and supported the interpretation of findings during the trial's dissemination phase.

The topic guide was developed with two members of the public, who helped guide the approach and style of the interview topic guides through a pilot interview (online supplemental file 1). Learnings from the pilot included some rewording of questions for clarity and the use of prompts to further unpick experiences. As suggested by patient and public involvement members, the topic guide was shared with participants prior to the interview, enabling them to reflect and plan beforehand.

### FINDINGS
### Participants
In total, 10 interviews were undertaken with caregiver dyads (20 participants) across our 5 study sites in England. One dyad declined due to other commitments, one did not give a reason and the remaining dyads did not take part in the interview as the person with hip fracture had been readmitted to hospital. The median length of interviews was 33 min (range: 27–53 min).

Dyad characteristics are presented in table 1. The median age of people with hip fracture and their caregiver was 72.5 years (range: 65–96 years) and 71.0 years (range: 43–81 years), respectively. Of the informal caregivers, seven were male and three were female. There were three males and seven females who experienced a hip fracture. Overall, six informal caregivers were the spouse of the person with hip fracture, two were adult children (a daughter and a son) and two were described as 'other'.

### Themes
Overall, two main themes and associated subthemes were identified:

### Expectations of the informal caregiver role
a. Expectations of care and services.
b. Responsibility and advocacy.

### Reality of being an informal caregiver
a. Profile of people with hip fracture.
b. Decision to be a caregiver.
c. Transition from hospital to home.

### Expectations of the informal caregiver role
Informal caregivers perceived that health professionals assumed that they would immediately take-on (additional) caregiving responsibilities on the discharge home.

> 'Assumed that because I'm working from home (I would care), just said it like, didn't even ask me.' (Caregiver, Female, Control Group)

### Expectations of care and services
Both informal caregivers and people with hip fracture suggested that rather than being personalised, goals were set by health professionals on a 'standard' patient recovery trajectory and therefore these were not viewed as achievable, nor realistic. They also had insufficient information to understand what a typical recovery trajectory looks like. It appears that most people did not feel empowered and found goal setting and developing realistic recovery goals difficult, based on imperfect information.

> 'I had no expectations because I have never been in hospital before. Ever. I have never been ill before, so this is completely new to me and obviously new to (caregiver)—he's never had to care for me before.' (Person with Hip Fracture, Female, Intervention Group)

> 'Not quite enough available to you about kind of next steps and what's going to happen in the future.' (Caregiver, Male, Control Group)

In contrast, again for informal caregivers and people with hip fracture, receiving knowledge on recovery expectations with realistic goal-setting and joint decision-making, as seen in our HIP HELPER intervention, appeared to be empowering.

> 'If you get somebody who doesn't have sufficiently stimulating support, that is a problem because in a sense, it encourages them to stay dependent, whereas what it should be doing is encourage them to become independent.' (Caregiver, Male, Intervention Group)

### Responsibility and advocacy
Caregivers reported a sense that they must advocate and even battle for services to achieve adequate care and support in the transition from hospital to home, and yet, they had no power to do so. Navigating the care system is difficult and requires knowledge and confidence. This was particularly pertinent among first-time caregivers.

> 'There is only as much as I can do to protect my husband. He's got a right to have a minimum care. Really, we're not asking for much.' (Caregiver, Female, Intervention Group)

**Table 1** Characteristics of sample interviewed

| | Intervention group N=7 | Control group N=3 | Overall N=10 |
|---|---|---|---|
| **Informal caregiver** | | | |
| Age (years): median (IQR) | 71.0 (58.0–81.0) | 71.0 (43.0–72.0) | 71.0 (58.0–77.0) |
| Gender: n (%) | | | |
| Male | 5 (71.4) | 2 (66.7) | 7 (70.0) |
| Female | 2 (28.6) | 1 (33.3) | 3 (30.0) |
| Ethnicity: n (%) | | | |
| White British | 6 (85.7) | 3 (100) | 9 (90.0) |
| White other | 1 (14.3) | 0 | 1 (10.0) |
| Relationship to person with hip fracture: n (%) | | | |
| Spouse | 4 (57.1) | 2 (66.7) | 6 (60.0) |
| Daughter/son | 1 (14.3) | 1 (33.3) | 2 (20.0) |
| Other | 2 (28.6) | 0 | 2 (20.0) |
| Employment: n (%) | | | |
| Not working | 6 (85.7) | 3 (100) | 9 (90.0) |
| Part-time work | 1 (14.3) | 0 | 1 (10.0) |
| **Person with hip fracture** | | | |
| Age at consent (years): median (IQR) | 79.0 (70.0–82.0) | 69.0 (68.0–71.0) | 72.5 (69.0–79.0) |
| Gender: n (%) | | | |
| Male | 1 (14.3) | 0 | 1 (10.0) |
| Female | 6 (85.7) | 3 (100) | 9 (90.0) |
| Ethnicity: n (%) | | | |
| White British | 7 (100) | 3 (100) | 10 (100) |
| Has cognitive impairment (based on AMTS category): n (%) | 1 (14.3) | 0 | 1 (10.0) |
| AMTS score at consent: median (IQR) | 10.0 (9.0–10.0) | 10.0 (10.0–10.0) | 10.0 (9.0–10.0) |
| NEADL score at baseline: median (IQR) | 20.0 (14.0–22.0) | 10.0 (10.0–10.0) | 17.0 (12.0–22.0) |
| **Location** | | | |
| Site (n) | | | |
| 1 | 1 | 1 | |
| 2 | 1 | 1 | |
| 3 | 1 | 1 | |
| 4 | 3 | – | |
| 5 | 1 | – | |

AMTS, Abbreviated Mental Test Score; NEADL, Nottingham Extended Activities of Daily Living scale.

'I had to make phone calls and chase around to try and sort things out (for her recovery at home).' (Caregiver, Female, Control Group)

Informal caregivers felt that this level of advocacy was time-consuming and created an imbalance in undertaking care activities, thus detracting from giving personal support.

'But while I have to chase up all these things, he's not getting the best. He needs my attention, you know, to make sure that he's doing his exercise.' (Caregiver, Female, Intervention Group)

Caregivers also expressed their feelings of accountability for supporting independence and recovery and, not surprisingly for some, this induced feelings of stress and anxiety.

'She's always been very independent, and I just felt very responsible for it. Seems to be quite a lot on my shoulders.' (Caregiver, Male, Intervention Group)

'There was a shock when she fell and broke her hip. It's a lot to take on. All of a sudden, everything falls to one person, it's tough.' (Caregiver, Male, Control Group)

### Reality of being an informal caregiver

There was a dissonance in the expectations of being a caregiver and the actual role. The reality was shaped by the motivation and willingness to become a caregiver, the profile of the caregiver and the person with hip fracture, and the process of being discharged home. This mismatch was seen more in first-time caregivers and spouses of the person with hip fracture.

### Profile of the caregiver and the person with hip fracture

The majority (n=8/10) of caregivers were first-time caregivers with little or no understanding of the hip fracture recovery process.

'His wife had Alzheimer's and he cared for her for about 16years.So,it's something that he understands. And he,well,he's just very good at it.' (Person with Hip Fracture,Female,Intervention Group)

For those caregivers in full-time employment, competing priorities between work and caring highlighted the reality of being an informal caregiver.

'I had loads of work on and I was having to work at 10 o'clock at night to try and get my work done,because through the day when places were open,I was fooling around trying to get the things we need,sort mam and do her appointments.' (Caregiver,Female,Control Group)

The profile of people with hip fracture in this study was generally active older adults, some of whom were still in employment and felt the services available post-hip fracture were not conducive to their lifestyle, needs or goals. They perceived services are primarily designed for older adults, suggesting inadequacy in rehabilitation for those with hip fracture (and their caregivers) aiming to return to occupational or physically demanding leisure activities.

'I don't think we're typical patients and carers. Certainly,I am definitely not a typical carer,and *** is definitely not a typical patient.' (Caregiver,Male,Intervention Group)

'You know,I'm not a little old female who sits and knits in the chair and doesn't move about much so that makes a difference,doesn't it?' (Person with Hip Fracture,Female,Intervention Group)

Caregivers were also mindful of their own age and health status. Caregivers were aware that they required physical and mental capacity to effectively care for the person following hip fracture and that maintaining their own health and well-being was equally as important.

'I was conscious that I had to take care of myself to be fit to be looking after her.' (Caregiver,Male,Intervention Group)

'You know I'm so lucky that I'm healthy still.That I'm young and capable,but,who knows,you know?' (Caregiver,Male,Intervention Group)

Caregivers recognised the need to balance their own physical and psychological needs with their care recipient's need.

### Decision to be a caregiver

Findings identified multiple factors which contributed to the decision to become an informal caregiver. Spousal obligation and acting out of affection were examples identified from the present study. For some, values such as loyalty and commitment arising from marriage (spousal obligation) were what motivated people to take on a caregiving.

'We both met two and a half years ago. Dare I say it nuts about each other?Ever since then,we've been living together,as man and wife.' (Caregiver,Male,Intervention Group)

For others, there was the absence of actual choice, in-part attributed to assumptions made on discharge planning, but also underpinned by the unavailability of alternative care options.

'I think in our case there wouldn't have been anyone else who would be able to come and help.' (Caregiver,Male,Intervention Group)

An interesting point about the decision to be a caregiver is that no participant expressed an awareness of caring for a limited acute period of time. Thus, it is unknown how long caregivers perceive the responsibility of caring for a person with hip fracture lasts for.

### Transition from hospital to home

Caregivers expressed frustration, confusion and uncertainty at the point of transitioning home. Such feelings were principally attributed to a lack of communication throughout discharge planning with caregivers feeling isolated, underprepared and excluded from the decision-making. Visitor restrictions due to the COVID-19 pandemic may have also reduced communication channels and contributed to caregiver uncertainty, resulting in concerns of whether the person with hip fracture was receiving adequate.

'It's very difficult. I was at one point sending her back to hospital 'cause I was so frantic with it all.' (Caregiver,Female,Intervention Group)

'Nothing is said (about discharge)at the hospital,apart from telling you have an appointment ineight weeks,there's nothing else said.' (Caregiver,Male,Intervention Group)

'I mean the one thing that I do remember is really actually a feeling of isolation and a little bit concerned at whether or not I was doing the right things to support her.' (Caregiver,Male,Intervention Group)

The profile of caregivers also contributes to their competence in managing the transition home, potentially relying on previous experiences to navigate the health system and available services.

'You know,imagine if you if you weren't an old nurse like me? How on earth can you even navigate the system? I think people are getting lost out there and it's creating a lot of problems for the health service.'(Caregiver,Female,Intervention Group)

Some participants shared how they had to 'beg, steal and borrow' equipment from friends and family. It is apparent that the onus is on the caregiver themselves to navigate and seek the appropriate provisions for recovering at home with hip fracture and there was a feeling of abandonment.

'Luckily,some family had a spare single bed. A friend had a perching stool and a trolley.And then another friend,sadly her husband had been ill,but they had purchased their own wheelchair and we actually borrowed that wheelchair.And a cousin,he built some ramps so I could get the wheelchair in.'(Caregiver,Female,Control Group)

## DISCUSSION

Previous studies have provided important insights into the experiences of patients and caregivers following hip fracture, which we have further explored to better understand this challenge.[13 28] Our study aimed to understand the perspectives of informal caregivers who support the recovery of people following hip fracture surgery. Findings indicate that there is a tension between the expectations, and reality of the caregiving role for people with hip fracture. This is largely attributed to caregivers feeling disempowered to advocate for a person's recovery, leading to increased stress, anxiety and frustration when supporting them. Additionally, caregivers struggle with 'juggling' their own life with their caregiving role (eg, full-time employment),[29] while also maintaining their own health and well-being, for the benefit of both members of the dyad when discharged home.[30] There is a growing body of literature suggesting the mental health and wellness of a caregiver is also linked with a patient's functional and health outcomes.[30–32] Maintaining physical and mental ability to care for an individual is therefore important for hip fracture recovery.

This study identified that discharge processes after hip fracture do not currently fully prepare informal caregivers for their roles. As such, key learning points for health professionals are suggested to improve collaboration and support in the transition from hospital to home. These include: a greater explanation on 'normal' recovery processes; engaging for (dyad) joint decision-making; and promote the provision of additional (community) services for support. This is supported by previous findings that have also highlighted several key recommendations for translational care which emphasise dyad engagement, education and well-being.[33 34]

Joint decision-making based on setting realistic and tailored goals, is another practical strategy health professionals may employ to support the transition of people with hip fracture and their informal caregiver from hospital to home. This agrees with previous authors such as Angeli et al[35] and Brewer et al[36] and re-enforced in Saletti-Cuesta et al's[13] systematic review, who stressed that collaborative goal setting can allow designated time to enhance family involvement, to focus on functional outcomes and highlight current priorities for the caregiving dyad. It, therefore, may be that informal caregiver are well placed to act as a 'driver of goal-setting', as they may have intimate knowledge of the person pre-hip fracture (eg, preferences and character).

A key finding of this study was the powerlessness which caregivers felt they had in supporting and advocating for the person with hip fracture. As a result of their inability to advocate for the person with hip fracture, participants frequently reported feelings of either over anticipation (negative) contrasted with unrealistic (positive) opinions on what could be offered and provided to the person who they were supported. This is a common theme that resonates within previous literature.[18 37 38] Tutton et al[28] acknowledged the uncertainties shared by patients and caregivers after hip fractures and change of relationship within the caregiving dyad through disability. There was an expectation by health professionals did not sufficiently appreciate this uncertainty and particularly following a change in health status. Our findings suggest that health professionals expected that caregivers could navigate the health and social care system, which would indeed, require some level of health literacy. It is known that low health literacy among caregivers has the potential to impact adequate care provision.[39] This is particularly pertinent among those who are 'new' caregivers, those who may perceive to be particularly underskilled, underprepared and/or be adjusting to a new role or identify as an informal caregiver. This is contrary to experienced caregiver who could elicit competence, knowledge and tolerance in caregiving.[40–43] Additionally, the profile of a caregiver may include a natural affinity to caring, namely a 'caring nature'.[29 44] Therefore, there is the need to develop interventions that improve informal caregiver health literacy, ultimately upskilling informal caregivers to have the confidence to advocating for the person with hip fracture.

This study offers insights into an informal caregiver population which may differ to most previously reported. Previous literature has emphasised the caregiver experience of caring for people with long-term conditions which is unlikely to be transferable to the acute, traumatic and rapid nature of caregiving for people with hip fracture.[45] This study, therefore, offers unique perspectives on a different transition in becoming a (informal) caregiver, particularly as the findings suggest little time for decision-making and the assumption of accepting this role of significant responsibility.[46] A second element to this caregiving population is how it is gendered. Caregiving is known to have a greater adverse effect on female caregivers' mental health and life satisfaction.[47] However,

the demographic of hip fracture is largely female (considering experiences of the menopause and osteoporosis, to name a few), as seen in our larger study necessitating male partners to take on the caring role who may find undertaking sensitive tasks more challenging.[45 48] Additionally, the qualities and skills frequently offered by female caregivers previously reported are that females are better communicators, better instigators of help and are better organised, suggesting that male caregivers might benefit with tailored support for these potential challenges.[44] While it is recognised that expertise in managing frail, older patients is crucial to ensure holistic care for hip fracture patients,[49] our study also featured much younger hip fracture participants, who had quite different perceptions of what was important to them and expectations of recovery.[29 50]

This study presents strengths and limitations. A notable strength was its ability to capture experiences during a novel time in healthcare history, where hospitals were making rapid, visitor policy changes in response to the COVID-19 pandemic. Due to the timing of the study, the participants shared a distinctive experience and its effect on rehabilitation and on the caregiver experience. However, one may also interpret this as weakness as it is not a typical reflection of care for this population. Interviews took place within 4 months post-hip fracture, despite previous literature suggesting physical recovery may take up to 12 months,[51] and high levels of caregiver burden reported by caregivers at 6 months postsurgery.[37] There is currently little data regarding caregiver perspectives on the long-term recovery/care of hip fracture (eg, deconditioning and confidence in performing activities), thus understanding perspectives of the different caregiving trajectory, as people become more experienced carers and as people recover from hip fracture, would be valuable. Furthermore, it is unknown how long caregivers perceive the responsibility of caring for a person with hip fracture lasts for. No participant expressed an awareness of caring for an acute period of time and therefore how long this might last.[52]

It is likely that group allocation (ie, receiving the HIP HELPER intervention) may have impacted people's experiences. For example, the HIP HELPER workbook was designed to support goal-setting, thus such guidance may have influenced the way in-which recovering from, or caring for someone with a hip fracture may be perceived by participants. We acknowledge findings may not be fully generalisable because of the small sample size and England-based focus. Participants presented with limited socioeconomic, cultural or ethic perspectives, despite our best endeavours and there was also a lack of perspectives of those caring for a person with cognitive impairment.[53] This is important as such differences may be a source of variation in attitudes, behaviours and perspectives of caregiving for someone following hip fracture.[54 55] We also appreciate that the present cohort was largely of lower mean age than may be expected for a hip fracture population, however, this work provides more novel insights into the experiences of caregiving for younger people which is a profile anecdotally noted in practice.[56] Interviews were undertaken with informal caregivers and people with hip fracture together. This has limitations in that some individuals may be reluctant to address difficulties in the presence of the other person, some may assert dominance or reflect a particular status, and there is a likelihood of dyad interviews to be 'off-topic', compared with one-to-one interviews.[57]

## CONCLUSIONS

While there may be an expectation that informal caregivers will provide support for people following hip fracture, our study illuminates challenges faced due to the acute and traumatic nature of hip fracture. We suggest that joint decision-making and goal setting with health professionals, patients and their informal carers will enable better preparedness for the transition from hospital to home. This includes a greater explanation and tailoring of the recovery pathway especially for younger hip fracture patients.

**Author affiliations**
[1]School of Health Sciences, University of East Anglia, Norwich, UK
[2]Department of Clinical Gerontology and Geriatric Rehabilitation, Robert Bosch Hospital, Stuttgart, Germany
[3]Norwich Clinical Trials Unit, University of East Anglia, Norwich, UK
[4]Norwich Medical School, University of East Anglia, Norwich, UK
[5]Nuffield Department of Orthopaedics, Rheumatology and Musculoskeletal Sciences, University of Oxford, Oxford, UK
[6]Community Health Sciences, University of Nottingham, Nottingham, UK
[7]Rehabilitation, Aged and Extended Care, Flinders University, Adelaide, South Australia, Australia
[8]College of Medicine and Health, University of Exeter, Exeter, UK
[9]University of Warwick, Coventry, UK

**Collaborators** The HIP HELPER Study Collaborators: Penny Clifford (Norfolk, PPI Representative), Lis Freeman (Norfolk, PPI Representative), Rene Gray (Principal Investigator – James Paget University Hospital NHS Trust), Yan Cunningham (Principal Investigator – City Hospitals Sunderland NHS Foundation Trust), Sarah Langford (Principal Investigator - Northumbria Healthcare NHS Foundation Trust), Dr Mark Baxter (Principal Investigator - University Hospital Southampton NHS Foundation Trust), Jessica Pawson – (Principal Investigator - Barts Health NHS Trust), Melissa Taylor (James Paget University Hospital NHS Trust), Anna Mellows (James Paget University Hospital NHS Trust), Kate Lacey (James Paget University Hospital NHS Trust), Alex Herring (City Hospital Sunderland NHS Foundation Trust), Diane Williams (Northumbria Healthcare NHS Foundation Trust), Anna Cromie (Northumbria Healthcare NHS Foundation Trust), Gail Menton (Northumbria Healthcare NHS Foundation Trust), Warren Corbett (University Hospital Southampton NHS Foundation Trust), Helen Jowett (University Hospital Southampton NHS Foundation Trust), Vishwanath Joshi (Barts Health NHS Trust), Maninderpal Matharu (Barts Health NHS Trust), Maria Baggot (University Hospital Southampton NHS Foundation Trust) and David Barker (University Hospital Southampton NHS Foundation Trust).

**Contributors** SHa, AW, TS, SHo, KP, RK, P-AA, PL, MC, SL and MLC researched the topic and devised the study. TS, SHa, RK, P-AA, ABC, SHo, AW, KP, PL, MLC, SL and MC provided the first draft of the manuscript. SHa lead the qualitative study design. AW lead the qualitative research conduct. TS, SHa, AW, RK, P-AA, ABC, SHo, KP, PL, MC, SL and MLC contributed equally to manuscript preparation. TS acts a guarantor.

**Funding** This project is funded by the National Institute for Health and Care Research (NIHR) under its Research for Patient Benefit (RfPB) Programme (Grant Reference Number NIHR200731). MLC and SHo role in this study was supported

by the National Institute for Health and Care Research Oxford Biomedical Research Centre. SL role in this study was supported by the National Institute for Health and Care Research Exeter Biomedical Research Centre.

**Disclaimer** MCo and SHo are supported by the National Institute for Health Research (NIHR) Oxford Biomedical Research Centre (BRC). SEL is supported by the National Institute for Health and Care Research Exeter Biomedical Research Centre. The views expressed are those of the author(s) and not necessarily those of the NHS, the NIHR or the Department of Health and Social Care.

**Competing interests** None declared.

**Patient and public involvement** Patients and/or the public were involved in the design, or conduct, or reporting, or dissemination plans of this research. Refer to the Methods section for further details.

**Patient consent for publication** Consent obtained directly from patient(s).

**Ethics approval** This study involves human participants and was approved by North East - Newcastle and North Tyneside 1 Research Ethics Committee (20/NE/0213). Participants gave informed consent to participate in the study before taking part.

**Provenance and peer review** Not commissioned; externally peer reviewed.

**Data availability statement** Data are available on reasonable request. The data that support the findings of this study are available from the corresponding author (AW) on reasonable request. This includes access to the full protocol, and anonymised participant-level dataset.

**ORCID iDs**
Allie Welsh http://orcid.org/0000-0001-8278-6673
Sarah Hanson http://orcid.org/0000-0003-4751-8248
Allan Clark http://orcid.org/0000-0003-2965-8941
Sally Hopewell http://orcid.org/0000-0002-6881-6984
Pip Logan http://orcid.org/0000-0002-6657-2381
Maria Crotty http://orcid.org/0000-0002-2996-5135
Matthew Costa http://orcid.org/0000-0003-3644-1388
Toby Smith http://orcid.org/0000-0003-1673-2954

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
