## [Reviewer comments · BMJ Open]

ARTICLE DETAILS

TITLE (PROVISIONAL)	Perspectives of informal caregivers who support people following hip fracture surgery: a qualitative study embedded within the HIP HELPER feasibility trial
AUTHORS	Welsh, Allie; Hanson, Sarah; Pfeiffer, K; Khoury, Reema; Clark, Allan; Ashford, Polly-Anna; Hopewell, Sally; Logan, Phillipa; Crotty, Maria; Costa, Matthew; Lamb, Sarah; Smith, Toby; HIP HELPER Study, Collaborators

VERSION 1 – REVIEW

REVIEWER	Snowdon, David Monash University, Peninsula Clinical School
REVIEW RETURNED	26-Jun-2023

GENERAL COMMENTS	Thank you for the opportunity to review your paper on perspectives of informal caregivers who support people following hip fracture surgery. The findings would be of interest to researchers and clinicians with an interest in hip fracture rehabilitation/care, and consumers of hip fracture rehabilitation/care. I have some suggestions below to improve the quality of your manuscript. Introduction:  - On page 5 line 35 you state: 'Whilst caregiving experiences have been reported in other populations such as stroke [6] and dementia [11], there is still much to understand about acute episodes of informal care, such as in hip fracture.' It would be worthwhile referring to the previous literature exploring caregiver experiences of providing informal care following hip fracture. For example, this systematic review: Lorena Saletti-Cuesta, Elizabeth Tutton, Debbie Langstaff & Keith Willett (2018) Understanding informal carers' experiences of caring for older people with a hip fracture: a systematic review of qualitative studies, Disability and Rehabilitation, 40:7, 740-750, DOI: 10.1080/09638288.2016.1262467. This would help to better highlight the gap in the literature that your study addresses. Methods:  - Stating the inclusion criteria for the main trial would give the reader a better understanding of the larger cohort (of patients with hip fracture and informal carers) that the 20 participants in this qualitative study are representing. - Data analysis: thematic analysis is not a 'one size fits all' approach (refer to Virginia Braun & Victoria Clarke (2021) One size fits all? What counts as quality practice in (reflexive) thematic analysis?, Qualitative Research in Psychology, 18:3,328-352, DOI: 10.1080/14780887.2020.1769238). Therefore, could you please
------------------	--

	provide further detail on your qualitative analysis? For example, were data coded? Who coded data?  - Data analysis: please provide some information on team reflexivity and any reflexivity exercises that were undertaken prior to or during data analysis. Results:  - On page 6 line 42 you state that seven participants identified as a spouse, yet in Table 1 six participants identified as a spouse. Discussion:  - Please refer to previous literature on informal caregivers experiences of caring for people with hip fractures (i.e. Lorena Saletti-Cuesta, Elizabeth Tutton, Debbie Langstaff & Keith Willett (2018) Understanding informal carers' experiences of caring for older people with a hip fracture: a systematic review of qualitative studies, Disability and Rehabilitation, 40:7, 740-750, DOI: 10.1080/09638288.2016.1262467) and highlight/explain what your study adds to this literature. - On page 10 line 38 you state: 'A previous study by Tutton et al (31) has provided important insights into the experiences of patients and caregivers following hip fracture, which we have further explored to better understand this challenge', however, the study referenced explored experiences of patients and caregivers following stroke. You may have referenced the wrong article. Can you also please explain what the Tutton article found and what your study adds to the findings of the Tutton study (as per my previous comment).
--	--

REVIEWER	Könsgen, Nadja University Witten Herdecke Faculty of Health, Institute for Research in Operative Medicine
REVIEW RETURNED	03-Aug-2023

GENERAL COMMENTS	I would like to thank the editors for giving me the opportunity to review the manuscript. Of course, many thanks to the authors for their elaborate work. There are some very interesting aspects in the manuscript. However, I would like to speak out in favor of significantly revising the manuscript in some parts (especially in the results section). I divided my comments into major and minor comments: Major comments Methods  • “The qualitative sub-study aimed to explore the experiences and contextual factors of participating in HIP HELPER and their experiences of giving and receiving informal care” (p.6, ll.3-6)  Experiences and contextual factors related to participation in HIP HELPER play a very important role in the interview topic guide. However, I do not see this clearly reflected in the results. Please add experiences to participation to the results. In my comments on the results section, I have added suggestions where you could go into more detail on this. For example, you could mention if certain topics were only mentioned by people from the intervention or the control group. However, it would then make sense to elaborate on what exactly the intervention entailed. This refers in particular to what support is provided for informal caregivers in the intervention.
------------------	--

- The results section should be revised. In some parts, sentences are inserted that go beyond a description of the results and interpret them or put them in context with other literature. These should be moved to the discussion. Examples are (but there are more): “Such findings are comparable to literature exploring caregiver perceptions of caring for a person with stroke, as there is an immediate demand to assume a role of significant responsibility [18]” (p.7, ll. 9-11) and “This is an important area for discussion as previous literature suggests that caregivers who are in employment may experience constrained motivations to care, resulting in a source of tension [26]” (p.8, ll-39-42).

Minor comments Introduction:

- „Whilst caregiving experiences have been reported in other populations such as stroke [6] and dementia [11], there is still much to understand about acute episodes of informal care, such as in hip fracture“ (p. 5, ll.35-37)  Please describe in more detail why the results are not transferable and why a separate analysis is needed for patients with hip fractures. Methods
- “Caregiver dyads who were randomized to receive the HIP HELPER intervention were allocated to received three, one- hour, one-to-one training sessions, delivered by a nurse, physiotherapist, or occupational therapist“ (p. 5, ll.51-54)  Does each participant receive a session delivered by a nurse, one session delivered by a physiotherapist and one by a occupational therapist? • Please specify if participants were involved in data processing or analysis.
- Did the piloting result in any modifications?
- Was data analysis performed by the interviewer? By a single author? Or by more than 1 author?  If more than 2 authors analyzed the data, please specify how the calibration proceeded. If only 1 author analysed data or if there was no calibration process, please add this to the limitation section.
- Please add a statement on whether you reached data saturation.

Results • How long did the interviews take? Please provide median and range.

- “The mean age of people with hip fracture and their caregiver was 72.5 years (range: 65 to 96 years) and 71.0 years (range: 43 to 81 years), respectively“ (p.6, ll.39-41) Due to the small sample and the sensitivity of the mean to outliers, I suggest reporting the median.
- “Overall, seven informal caregivers were the spouse of the person with hip fracture, two were adult children (a daughter and a son) and two were described as ‘other’“ (p.6, ll.43-45)  In total, this results in 11 informal caregivers, but above it is listed that there were 10 dyads. Please correct.
- There were 7 dyads from the intervention and 3 from the control group. Could this have any impact on your results? Please add this to the discussion section.
- Please add to the section “Expectations of the informal caregiver role” which perspectives are included. That of the informal caregiver? The persons with hip fracture? The health professionals?
- “It may therefore be understandable that” (p.7, l. 22) is judging and should be formulated neutrally.

	 • The quote “If you get somebody who doesn’t have sufficiently stimulating support, that is a problem because in a sense, it encourages them to stay dependent, whereas what it should be doing is encourage them to become independent” (p. 7, ll. 37-39) could be better placed in context. I would rather have expected a quote here about what is so helpful about the HIP programme (as a kind of solution approach). Perhaps you could go into a little more detail here. • “Caregivers reported a sense that they must advocate and even battle for services to achieve adequate care and support in the transition from hospital to home, and yet, they had no power to do so” (p. 7, ll.43-45)  Was this also the case for interviews in the intervention group? Again, it might help to elaborate a little more on the intervention and whether other issues were raised by the participants in the intervention group. • “The profile of people with hip fracture in this study, were generally active older adults, some of whom were still in employment and felt the services available post-hip fracture, were not conducive to their lifestyle, needs or goals“ (p.8, ll.48-51)  what about the participants in the intervention group? Were there any differences in topics? • In the section “Transition from hospital to home” you could again add some information on experiences with the intervention. Discussion  • If I understand this correctly, you have conducted your interviews with the informal caregivers and the people with hip fracture together. Could this have biased your results? Perhaps the informal caregivers were reluctant to address some difficulties in the presence of the person with hip fracture. • “Findings from the present study highlight the position of informal caregivers as ‘drivers of goal-setting’, as they may have intimate knowledge of the person pre-hip fracture (for example, preferences and character)“ (p. 11, ll.7-9)  Is this elaborated in the results? • “A notable strength was its ability to capture experiences during a novel time in healthcare history, where hospitals were making rapid, visitor policy changes in response to the COVID-19 pandemic. Due to the timing of the study, the participants shared a distinctive experience and its effect rehabilitation and on the caregiver experience“ (p.11, ll. 45-48)  I would rather classify this as a limitation, because it may cause different topics to be critically noted than usual. Otherwise, the reference to the special health care situation during COVID would have to be worked out more clearly during the whole manuscript. Furthermore, there is an “on” missing in “its effect rehabilitation”.
--	---

VERSION 1 – AUTHOR RESPONSE

Comments from Reviewer 1

On page 5 line 35 you state: 'Whilst caregiving experiences have been reported in other populations such as stroke [6] and dementia [11], there is still much to understand about acute episodes of informal care, such as in hip fracture.' It would be worthwhile referring to the previous literature exploring caregiver experiences of providing informal care following hip fracture. For example, this systematic review: Lorena Saletti-Cuesta, Elizabeth Tutton, Debbie Langstaff & Keith Willett (2018)

Understanding informal carers' experiences of caring for older people with a hip fracture: a systematic review of qualitative studies, *Disability and Rehabilitation*, 40:7, 740-750, DOI: 10.1080/09638288.2016.1262467. This would help to better highlight the gap in the literature that your study addresses.

Response:

Thank you. We have reviewed this helpful paper and acknowledged this in the Introduction as suggested by Reviewer 1 (Introduction, Paragraph 4, Lines 10-13).

Comments from Reviewer 1

Stating the inclusion criteria for the main trial would give the reader a better understanding of the larger cohort (of patients with hip fracture and informal carers) that the 20 participants in this qualitative study are representing.

Response:

We have now included in the inclusion criteria for the main trial within the Methods section (Methods, Paragraph 3, Lines 1-9).

Comments from Reviewer 1

Data analysis: thematic analysis is not a 'one size fits all' approach (refer to Virginia Braun & Victoria Clarke (2021) One size fits all? What counts as quality practice in (reflexive) thematic analysis?, *Qualitative Research in Psychology*, 18:3,328-352, DOI: 10.1080/14780887.2020.1769238). Therefore, could you please provide further detail on your qualitative analysis? For example, were data coded? Who coded data?

Response:

As suggested, we have now provided more detail on the thematic analysis (Methods, Data Analysis, Paragraph 1, Lines 1-4).

Comments from Reviewer 1

Data analysis: please provide some information on team reflexivity and any reflexivity exercises that were undertaken prior to or during data analysis.

Response:

We have now included notes on reflexivity practices in the =Methods section (Methods, Data Analysis, Paragraph 2, Lines 1-4).

Comments from Reviewer 1

On page 6 line 42 you state that seven participants identified as a spouse, yet in Table 1 six participants identified as a spouse.

Response:

This has now been corrected on page 6 (six participants identified as a spouse) (Findings, Paragraph 2, Line 4).

Comments from Reviewer 1

Please refer to previous literature on informal caregivers experiences of caring for people with hip fractures (i.e. Lorena Saletti-Cuesta, Elizabeth Tutton, Debbie Langstaff & Keith Willett (2018) Understanding informal carers' experiences of caring for older people with a hip fracture: a systematic review of qualitative studies, *Disability and Rehabilitation*, 40:7, 740-750, DOI: 10.1080/09638288.2016.1262467) and highlight/explain what your study adds to this literature.

Response:

We have provided reference to this valuable paper in the Introduction (Introduction, Paragraph 4, Lines 10-13) in addition to reflecting on how our findings compliment Saletti-Cuesta's findings in the Discussion (Discussion, Paragraph 1, Line 1; Discussion, Paragraph 3, Line 4).

Comments from Reviewer 1

On page 10 line 38 you state: 'A previous study by Tutton et al (31) has provided important insights into the experiences of patients and caregivers following hip fracture, which we have further explored to better understand this challenge', however, the study referenced explored experiences of patients and caregivers following stroke. You may have referenced the wrong article.

Response:

Thank you for bringing this to our attention – indeed, the wrong article was referenced and this has been amended (Discussion, Paragraph 1, Line 1).

Comments from Reviewer 1

Can you also please explain what the Tutton article found and what your study adds to the findings of the Tutton study (as per my previous comment).

Response:

We have now reflected on how our findings compliment Saletti-Cuesta's findings in the Discussion (Discussion, Paragraph 1, Line 1; Discussion, Paragraph 3, Line 4) and Tutton et al (2021) (Discussion, Paragraph 4, Lines 6-8).

Comments from reviewer 2

The qualitative sub-study aimed to explore the experiences and contextual factors of participating in HIP HELPER and their experiences of giving and receiving informal care" (p.6, ll.3-6)  Experiences and contextual factors related to participation in HIP HELPER play a very important role in the interview topic guide. However, I do not see this clearly reflected in the results. Please add experiences to participation to the results. In my comments on the results section, I have added suggestions where you could go into more detail on this. For example, you could mention if certain topics were only mentioned by people from the intervention or the control group. However, it would then make sense to elaborate on what exactly the intervention entailed. This refers in particular to what support is provided for informal caregivers in the intervention.

Response:

Thank you for this comment, it has been an important reflection in the development of this article. The main feasibility study manuscript reports the patient and caregiver's perspective of the experimental intervention (currently under review with BMJ Open). The purpose of this current paper is to provide important results on the wider experiences and contextual factors of recovery after hip fracture for patients and caregivers. Therefore we have reported the more inductive findings related to giving and receiving informal care, as opposed to the experiences of the intervention (or usual care for control participants), which are reported in the main trial. We have amended the overall aim of the qualitative study which we hope better supports the scope of this manuscript and therefore made this distinction clearer to the reader (Methods, Paragraph 2, Lines 1-6).

Comments from reviewer 2

The results section should be revised. In some parts, sentences are inserted that go beyond a description of the results and interpret them or put them in context with other literature. These should be moved to the discussion. Examples are (but there are more): "Such findings are comparable to literature exploring caregiver perceptions of caring for a person with stroke, as there is an immediate demand to assume a role of significant responsibility [18]" (p.7, ll. 9-11) and "This is an important area

for discussion as previous literature suggests that caregivers who are in employment may experience constrained motivations to care, resulting in a source of tension [26]" (p.8, ll-39-42).

Response:

We have addressed this important point. The results section has now been revised in line with your recommendation to only describe findings, as opposed to further interpreting them within the context of other literature. These have now been moved to discussion section (Findings, Expectations of the informal caregiver role, Paragraph 1; Findings, Responsibility and Advocacy, Paragraph 2, Lines 2-3; Findings, Profile of the caregiver and the person with hip fracture, Paragraph 1, Lines 2-4 and Paragraph 2, Line 2-4 Paragraph 4; and Paragraph 5, Lines 2-5; Findings, Decision to be a caregiver, Paragraph 1, Lines 1-2; Discussion Paragraph 1, Lines 10-14; Discussion, Paragraph 4, Lines 5-6 and Lines 14-17).

Comments from reviewer 2

Whilst caregiving experiences have been reported in other populations such as stroke [6] and dementia [11], there is still much to understand about acute episodes of informal care, such as in hip fracture" (p. 5, ll.35- 37)  Please describe in more detail why the results are not transferable and why a separate analysis is needed for patients with hip fractures.

Response:

This has been added as suggested (Introduction, Paragraph 4, Lines 10-13).

Comments from reviewer 2

Caregiver dyads who were randomized to receive the HIP HELPER intervention were allocated to received three, one- hour, one-to-one training sessions, delivered by a nurse, physiotherapist, or occupational therapist" (p. 5, ll.51-54)  Does each participant receive a session delivered by a nurse, one session delivered by a physiotherapist and one by a occupational therapist?

Response:

This sentence has been amended for clarity. Sessions were delivered by either a nurse, physio or occupational therapist (Methods, Paragraph 1, Line 7).

Comments from reviewer 2

Please specify if participants were involved in data processing or analysis

Response:

Although throughout this study, we had significant PPI contributions, participants were not involved in the data processing of analysis. This has been updated and documented in the manuscript (Methods, Data Analysis, Paragraph 4, Lines 2-4; Patient and Public Involvement, Paragraph 2, Lines 1-4).

Comments from reviewer 2

Did the piloting result in any modifications?

Response:

Piloting of the interview resulted in only minor adaptations to the topic guide. These included the rewording of some questions for clarity and the addition of prompts to further unpick dyad experiences. This was originally not reported in the present article due to word count limitations, however, this has now been included within the PPI section of the Methods section (Patient and Public Involvement, Paragraph 2, Lines 1-4).

Comments from reviewer 2

Was data analysis performed by the interviewer? By a single author? Or by more than 1 author?  If more than 2 authors analyzed the data, please specify how the calibration proceeded. If only 1 author analysed data or if there was no calibration process, please add this to the limitation section.

Response:

Data analysis was performed by the interviewer (AW). This has now been more clearly documents (Methods, Data Analysis, Paragraph 1, Lines 3-4).

Comments from reviewer 2

Please add a statement on whether you reached data saturation.

Response:

Saturation was not assessed in the study. The term saturation only applies to research used Grounded Theory methods (Charmaz, 2014), which employs theoretical sampling. In terms of sampling, we initially started with a purposive sampling approach people based on diversity in age, pre- fracture disability, and hospital locations. In the event, due to poor response, we took a more pragmatic approach which resulted in us interviewing 10 dyads. This has now been clearly stated in the Methods section (Methods, Data Analysis, Paragraph 3, Lines 1-3).

Comments from reviewer 2

How long did the interviews take? Please provide median and range.

Response:

This has now been included. (Findings, Paragraph 1, Line 4)

Comments from reviewer 2

“The mean age of people with hip fracture and their caregiver was 72.5 years (range: 65 to 96 years) and 71.0 years (range: 43 to 81 years), respectively“ (p.6, ll.39-41) Due to the small sample and the sensitivity of the mean to outliers, I suggest reporting the median.

Response:

The median has now been reported (Findings, Paragraph 2, Line 2).

Comments from reviewer 2

“Overall, seven informal caregivers were the spouse of the person with hip fracture, two were adult children (a daughter and a son) and two were described as ‘other“ (p.6, ll.43-45)  In total, this results in 11 informal caregivers, but above it is listed that there were 10 dyads. Please correct.

Response:

Thank you for noticing this error, it has been corrected (Findings, Paragraph 2, Line 4).

Comments from reviewer 2

There were 7 dyads from the intervention and 3 from the control group. Could this have any impact on your results? Please add this to the discussion section.

Response:

Although we acknowledge the potential impact of the HIP HELPER intervention, discussing the difference between groups was not the scope of this study. Differences in experiences are noted within the qualitative sub-study of the HIP HELPER main trial (under review). This study aimed to only present the the experiences and contextual factors of undergoing hip fracture surgery and dyad's

experiences of giving and receiving informal care. The reporting of these particular results in this paper has now been more clearly demarcated in the paper (Methods, Paragraph 2, Lines 3-6).

Comments from reviewer 2

Please add to the section “Expectations of the informal caregiver role” which perspectives are included. That of the informal caregiver? The persons with hip fracture? The health professionals?

Response:

The perspectives of people with hip fracture, informal caregivers and health care professionals are presented. Sentences have been revised to ensure it is clear who’s perspective is presented (Findings, Expectations of care and services, Paragraph 1, Line 1 and Paragraph 2, Line 1).

Comments from reviewer 2

“It may therefore be understandable that” (p.7, l. 22) is judging and should be formulated neutrally.

Response:

This sentence has now been adjusted to clarify this point (Findings, Expectations of care and services, Paragraph 2, Line 1).

Comments from reviewer 2

The quote “If you get somebody who doesn’t have sufficiently stimulating support, that is a problem because in a sense, it encourages them to stay dependent, whereas what it should be doing is encourage them to become independent” (p. 7, ll. 37-39) could be better placed in context. I would rather have expected a quote here about what is so helpful about the HIP programme (as a kind of solution approach). Perhaps you could go into a little more detail here. “Caregivers reported a sense that they must advocate and even battle for services to achieve adequate care and support in the transition from hospital to home, and yet, they had no power to do so” (p. 7, ll.43-45) → Was this also the case for interviews in the intervention group? Again, it might help to elaborate a little more on the intervention and whether other issues were raised by the participants in the intervention group. “The profile of people with hip fracture in this study, were generally active older adults, some of whom were still in employment and felt the services available post-hip fracture, were not conducive to their lifestyle, needs or goals” (p.8, ll.48-51)  what about the participants in the intervention group? Were there any differences in topics? In the section “Transition from hospital to home” you could again add some information on experiences with the intervention.

Response:

It was not the intention of the present manuscript to present differences between the intervention and usual care groups. The aim and methods have been amended to reflect this (Methods, Paragraph 2, Lines 3-6).

Comments from reviewer 2

If I understand this correctly, you have conducted your interviews with the informal caregivers and the people with hip fracture together. Could this have biased your results? Perhaps the informal caregivers were reluctant to address some difficulties in the presence of the person with hip fracture.

Response:

Yes, that is correct, we undertook the interviews with informal caregivers and the people with hip fracture together. Thank you for bringing this to our attention – we have now reflected on this in the limitations section of the manuscript (Discussion, Paragraph 6, Lines 22-25).

Comments from reviewer 2

“Findings from the present study highlight the position of informal caregivers as ‘drivers of goalsetting’, as they may have intimate knowledge of the person pre-hip fracture (for example, preferences and character)” (p. 11, ll.7-9)  Is this elaborated in the results?

Response:

On reflection, this is not clear within the results and takes precedent within the main HIPHELPER study. We have now slightly reframed the the discussion point to reflect the results better (Discussion, Paragraph 3, Lines 6-9).

Comments from reviewer 2

“A notable strength was its ability to capture experiences during a novel time in healthcare history, where hospitals were making rapid, visitor policy changes in response to the COVID-19 pandemic. Due to the timing of the study, the participants shared a distinctive experience and its effect rehabilitation and on the caregiver experience” (p.11, ll. 45-48)  I would rather classify this as a limitation, because it may cause different topics to be critically noted than usual. Otherwise, the reference to the special health care situation during COVID would have to be worked out more clearly during the whole manuscript. Furthermore, there is an “on” missing in “its effect rehabilitation”.

Response:

Thank you for sharing your reflections on the limitations of undertaking this study during the Pandemic. We have now included this (Discussion, Paragraph 6, Lines 4-6). We have also corrected the typo.

VERSION 2 – REVIEW

REVIEWER	Snowdon, David Monash University, Peninsula Clinical School
REVIEW RETURNED	14-Sep-2023

GENERAL COMMENTS	Thank you for addressing my comments.
------------------	---------------------------------------

REVIEWER	Könsgen, Nadja University Witten Herdecke Faculty of Health, Institute for Research in Operative Medicine
REVIEW RETURNED	20-Sep-2023

GENERAL COMMENTS	The authors have done a good job of implementing the comments. I only have a few minor points to make. -“Sessions were delivered by either a nurse, physio or occupational therapist” (Methods, Paragraph 1, Line 7)  Thanks for your specification, but this was not my problem. My question was whether each patient had one session with a member of each professional group. Or was it also possible to have 2 sessions with
	a nurse and one with a physio (and none with an occupational therapist)? -Methods, Paragraph 3: Other inclusion criteria for participants were men and women 60 years and above, community-dwelling prior to admission, able to attend face-to-face hospital appointments and/or access to a computer/table and internet services to receive a video consultation call. Please specify to whom these criteria apply. The age, for example, only applies to the patients, if I understand correctly. -My original comment: "There were 7 dyads from the intervention and 3 from the control group. Could this have any impact on your results? Please add this to the discussion section." Thank you for the specification, I understood that evaluation or differences between groups was not the aim of your qualitative study. Nevertheless, I think it should be pointed out in the limitations that for some topics, for example transition from hospital to home, the themes might be influenced by the intervention. In view of the unequal distribution among the different groups, I find this very relevant. As you described, the HIP Helper Workbook provides information on, for example, goal-setting plans to facilitate a good recovery. Such guidance can have an impact on how the process is perceived.

VERSION 2 – AUTHOR RESPONSE

Comment: “Sessions were delivered by either a nurse, physio or occupational therapist” (Methods, Paragraph 1, Line 7)  Thanks for your specification, but this was not my problem. My question was whether each patient had one session with a member of each professional group. Or was it also possible to have 2 sessions with a nurse and one with a physio (and none with an occupational therapist)?

Response:

We have now revised this sentence for clarity (Methods (page 4), paragraph 2, line 6).

Comment:

Methods, Paragraph 3: Other inclusion criteria for participants were men and women 60 years and above, community-dwelling prior to admission, able to attend face-to-face hospital appointments and/or access to a computer/table and internet services to receive a video consultation call. Please specify to whom these criteria apply. The age, for example, only applies to the patients, if I understand correctly.

Response:

Thank you for bringing this to our attention. We have now revised the statement of the inclusion criteria to create distinction between people with hip fracture and the caregivers. (Methods (page 4), paragraph 4, line 2, 6).

Comment:

My original comment: "There were 7 dyads from the intervention and 3 from the control group. Could this have any impact on your results? Please add this to the discussion section." Thank you for the specification, I understood that evaluation or differences between groups was not the aim of your qualitative study. Nevertheless, I think it should be pointed out in the limitations that for some topics, for example transition from hospital to home, the themes might be influenced by the intervention. In view of the unequal distribution among the different groups, I find this very relevant. As you described, the HIP Helper Workbook provides information on, for example, goalsetting plans to facilitate a good recovery. Such guidance can have an impact on how the process is perceived.

Response:

We have now included which intervention group participants were allocated to in the findings and added this as a limitation in the discussion section. (Discussion, (page 11), paragraph 2, lines 1-4).